# Tumor Semantic Segmentation in Hyperspectral Images using Deep Learning

**Stojan Trajanovski**[1]                                         STOJAN.TRAJANOVSKI@PHILIPS.COM
**Caifeng Shan**[1]                                                   CAIFENG.SHAN@PHILIPS.COM
[1] *Philips Research, Eindhoven, The Netherlands*

**Pim J. C. Weijtmans**[2]                                    P.J.C.WEIJTMANS@STUDENT.TUE.NL
[2] *Eindhoven University of Technology (TU/e), Eindhoven, The Netherlands*

**Susan G. Brouwer de Koning**[3]                        S.BROUWERDEKONING@NKI.NL
[3] *Netherlands Cancer Institute, Amsterdam, The Netherlands*

**Theo J. M. Ruers**[3,4]                                                   T.RUERS@NKI.NL
[4] *fMIRA Institute, University of Twente, Enschede, The Netherlands*

## Abstract

Real-time feedback based on hyperspectral images (HSI) to a surgeon can lead to a higher precision and additional insights compared to the standard techniques. To the best of our knowledge, deep learning with semantic segmentation utilizing both visual (VIS) and infrared channels (NIR) has never been exploited with the HSI data with human tumors. We propose using channels selection with U-Net deep neural network for tumor segmentation in hyperspectral images. The proposed method, based on bigger patches, accounts for bigger spatial context and achieves better results (average dice coefficient $0.89 \pm 0.07$ and area under the ROC-curve AUC $0.93 \pm 0.04$) than pixel-level spectral and structural approaches in a clinical data set with tongue squamous cell carcinoma. The importance of VIS channel for the performance is higher, but NIR contribution is non-negligible.

**Keywords:** Tumor Segmentation, Hyperspectral imaging, Deep Learning

## 1. Introduction

Treating tumors in tongue or other parts is generally done by removing the tumor parts surgically. Accurate segmentation of the tumor tissue from the healthy part is challenging. Moreover, there are no commonly accepted techniques that can provide real-time feedback to the doctor during the surgery.

Initially designed by NASA/JPL (Goetz, 2009) for remote sensing, hyperspectral imaging (HSI) has been successfully applied in archaeology, food quality, resource control and biomedicine (Lu and Fei, 2014). Due to its ease of use, simple hardware and low costs yet alone with computational power optimizations, HSI has become an emerging imaging modality for medical applications such as real-time feedback support of a surgeon. HSI is composed of hundreds of redundant color bands in a multi-megapixel image, which makes it more challenging for processing compared to standard RGB data.

(Fei et al., 2017) have evaluated the use of HSI (450-900nm) on specimen from patients with head and neck cancer. They achieved an area under the ROC-curve (AUC) of 0.94 for tumor classification with a linear discriminant analysis on a data set of 16 patients. (Halicek

et al., 2017) applied convolutional network with leaving-one-patient-out cross-validation and achieved an accuracy of 77% on specimen from 50 head and neck cancer patients in the same spectral range. An animal study by (Ma et al., 2017) achieved an accuracy of 91.36% using convolutional neural networks. The specimen were taken from mice with induced tumors. In all the mentioned studies the focus is entirely on spectral information.

On a clinical data of 14 patients, the semantic segmentation (Ronneberger et al., 2015) with channel selection (Weijtmans et al., 2019) of both visual (VIS) and near-infrared channels (NIR) using 100 random patches per patient achieves the best performance of average dice coefficient and area under the ROC-curve of $0.89 \pm 0.07$ and $0.93 \pm 0.04$, respectively, in the leave-patients-out cross-validation setting. This performance is better than pixel-vise spectral and structural tumor detection (Weijtmans et al., 2019). Both VIS and NIR channels are important for performance, although VIS contribution is larger.

## 2. Clinical data set

Tissue of 14 patients with tongue squamous cell carcinoma has been resected by the surgeon. Directly after resection, the specimen was brought to the pathology department. The pathologist localized the tumor by palpation and cut the specimen in two parts, right through the middle of the tumor. HSI images are then taken from these new surfaces using a Specim HSI cameras (http://www.specim.fi/) in the visible range (400 - 1000nm) and near-Infrared range (900 - 1700nm). All processes performed at the hospital follow the ethical guidelines for ex vivo human studies. An RGB picture of the tissue is taken to function as an intermediate step in the registration process.

In order to label the HSI data, a histopatological slide is taken from the surface that has been scanned. The slide is digitized and delineated to mark the tumor (red), healthy muscle (green) and epithelium (blue), which is the first step in Figure 1(a). From the delineation a mask is created. During histopathological processing the specimen was deformed and to correct this, a non-rigid registration algorithm is used. Obvious matching points in the histopathological and RGB images were visually selected. Using these points, the mask is transformed to match the RGB picture. This is depicted in Figure 1(a) in middle row as transformation T1. The point-selection is done again on the RGB and HSI data to acquire transformation T2, which is used to transform the mask to match the HSI data.

## 3. Method and Results

Using the mask the data can be explored. Based on this we create patches for the HSI data cube and the annotation. The reason is two fold: (i) we have a limited data from 14 patients and in this way we create a decent train and validation set of patches that can lead to reasonable results and (ii) the spatial dimensions (length and width) of the HSI data cubes are different, thus some patching is anyway in order to have a unique input data shape for any neural network. To work with the limited data set, leave-two-patients-out cross validation is used during training. To have a balanced training process we take 100 random patches of size 256x256 for each patient with the central pixels being 50/50 tumor and healthy classes and appropriate channel selection of the 128 most significant channels based on (Weijtmans et al., 2019). This means that we use 1200 patches (12 patients)

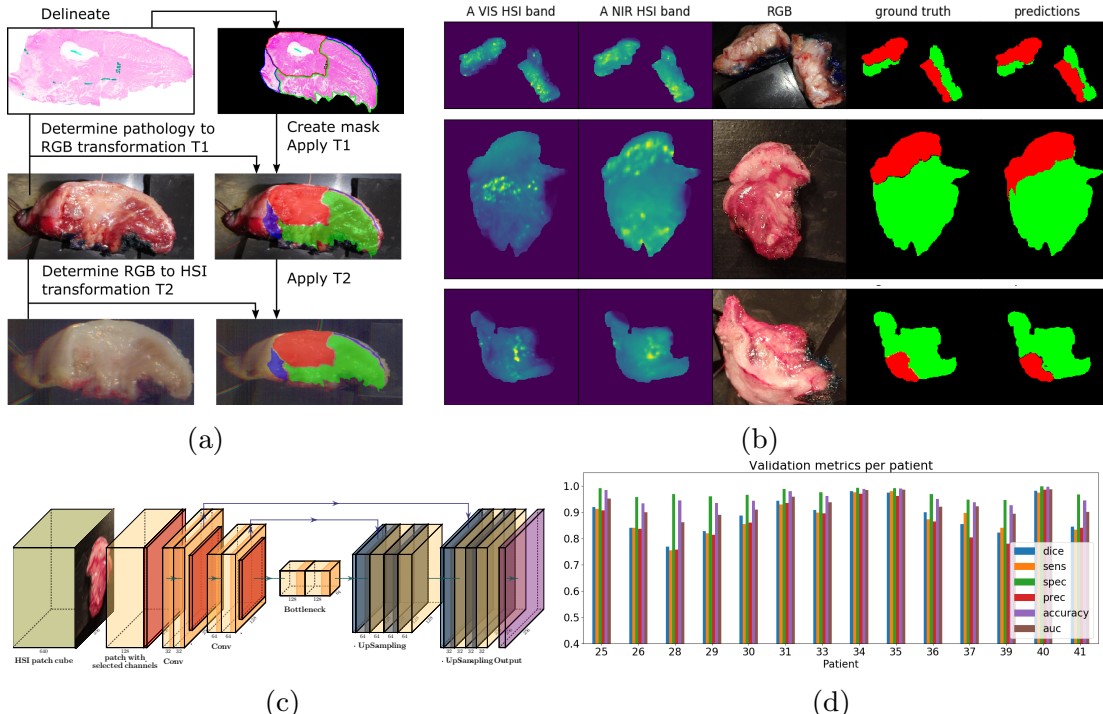

Figure 1: (a) Annotation of the hyperspectral data: tumor (red), healthy tongue muscle (green) and healthy epithelium (blue). (b) Ground truth and hard predictions of our model for 3 patients. (c) U-Net neural network architecture for HSI data (visualized using Plot-NeuralNet: https://github.com/HarisIqbal88/PlotNeuralNet). (d) Validation performance represented by the dice coefficient, sensitivity, specificity, precision, accuracy and AUC.

of size 256x256x128 for training and 200 patches (2 patients) for validating. Each patch contains both tumor and healthy tissue and in fact, covers significant spatial part of each HSI cube. To achieve better generalization, we apply standard generalization techniques such as patches rotation and flipping. Dice coefficient is used as a loss for training and validation that compares the overlap of the full size prediction map with the annotation map. With these input patches and annotations (size 256x256x2 tumor/no tumor), we train a U-Net neural network (Ronneberger et al., 2015) variant (Figure 1(c)). We use AUC, dice coefficient, sensitivity, specificity, precision and accuracy for evaluating the validation results. These give a reasonable indication for performance even in unbalanced data, without choosing a threshold for the classification. The validation results of these metrics per patient are given in Figure 1(d). The mean dice coefficient and AUC validation scores are 0.89±0.07 and 0.93±0.04, respectively, improving the performance of spectral and structural approaches (Weijtmans et al., 2019). To illustrate the accuracy of the prediction and the discrepancy with the ground truth labels, we depicted the hard predictions and the ground truth maps for 3 patients in Figure 1(b). The importance of VIS channels for performance is higher than the NIR counter parts, however NIR channels are crucial for spotting the tumor for some patients. Due to space constrains, this analysis is not presented.

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
