# OpenReview forum: "Tumor Semantic Segmentation in Hyperspectral Images using Deep Learning"
_MIDL.io/2019/Conference/Abstract — MIDL Abstract 2019_

### Official Review · AnonReviewer1 · 2019-04-24
**Multi-modal learning**

**Rating:** 3
**Confidence:** 3

**Review:**

This paper uses multi-modal data to learn tumor segmentation.
It is interesting to use both visual (VIS) and infrared channels (NIR) to segment human tumor.
The topic has strong clinical value.
The methodology is simple by using U-Net.

---

### Official Review · AnonReviewer2 · 2019-04-30
**CNN segmentation of the tumor in tongue using hyperspectral images**

**Rating:** 3
**Confidence:** 1

**Review:**

A U-net based analysis for tumor segmentation in the tongue using hyperspectral images is proposed.
The last paragraph of the introduction could be slightly rewritten to make a better distinction between the previous and current work, and also to better describe the task. Probably most of the MIDL audience – like me - is not familiar with the application, novelty or clinical relevance of the work.
Image analysis method is standard, but application seems to be novel. Experimental setting looks ok. Results seem solid, but they are difficult to interpret. The comparison is performed with prior work of one of the co-authors but results of that work are not given, nor is it clear whether the same data set is used.

---

### Decision · Program_Chairs · 2019-05-06
**Acceptance Decision**

Accept